# Harnessing Bacterial Agents to Modulate the Tumor Microenvironment and Enhance Cancer Immunotherapy

**DOI:** 10.3390/cancers16223810

**Published:** 2024-11-13

**Authors:** Christina James Thomas, Kaylee Delgado, Kamlesh Sawant, Jacob Roy, Udit Gupta, Carly Shaw Song, Rayansh Poojary, Paul de Figueiredo, Jianxun Song

**Affiliations:** 1Department of Microbial Pathogenesis and Immunology, Texas A&M University Health Science Center, MREB II, Room 3344, 8447 John Sharp Parkway, Bryan, TX 77807, USA; chrjamesthomas@tamu.edu (C.J.T.); kaylee_delgado@tamu.edu (K.D.);; 2Department of Molecular Microbiology and Immunology, The University of Missouri School of Medicine, Columbia, MO 65212, USA

**Keywords:** cancer immunotherapy, tumor microenvironment, engineered bacteria, immunosuppression, immune checkpoint inhibitors, bacterial therapeutics, antitumor immunity

## Abstract

This review explores the potential of engineered bacteria as a novel approach to cancer therapy. These microscopic organisms can target and infiltrate tumors, delivering medications and activating the body’s immune system to fight cancer cells. Early research shows promise, but more studies are needed to ensure safety and efficacy. If successful, this approach could lead to new treatments that are less harmful and more effective for many people with cancer. By understanding how these bacteria work and overcoming potential challenges, it may be possible to develop powerful new tools in the fight against this disease.

## 1. Introduction

Immunotherapy has revolutionized cancer treatment by harnessing the body’s immune system to identify and eliminate tumor cells. Unlike traditional treatments like chemotherapy and radiation, which directly target cancer cells, immunotherapy empowers the immune system to recognize and attack tumors more effectively [1]. However, the success of immunotherapy is often impeded by the tumor microenvironment (TME), a complex and hostile setting that suppresses immune responses.

The TME is a sophisticated network of tumor, stromal, and various immune cells collaborating to create an immunosuppressive environment [2]. Tumor cells within the TME evade immune detection and suppress antitumor immunity through mechanisms such as cytokine secretion, recruitment of regulatory immune cells, and expression of checkpoint molecules that inhibit immune cell function [3]. This interaction supports tumor growth and progression and contributes to resistance against immunotherapy.

Cancer cells actively shape the TME to support their survival and proliferation. They manipulate stromal cells and immune cells within the TME to create a favorable environment for tumor growth, inducing angiogenesis and remodeling the extracellular matrix to facilitate invasion and metastasis [4]. Furthermore, the TME is highly adaptable and responds to various intrinsic and extrinsic stresses, complicating efforts to target tumors effectively [5].

Given the critical role of the TME in cancer progression and immune evasion, developing strategies to alter its immunosuppressive nature is essential for improving immunotherapy outcomes. One innovative approach involves the use of engineered bacteria as therapeutic agents. Bacteria can be modified to specifically target and modulate the TME, enhancing the immune system’s ability to recognize and destroy tumor cells [6]. These bacterial therapies can deliver anti-cancer agents directly to the tumor site, produce immunostimulatory molecules, and disrupt the tumor’s protective environment, making them a promising adjunct to existing immunotherapy regimens.

This review delves into how bacterial agents and components affect immune cells and contribute to altering the TME. We explore different types of bacteria currently being used in therapeutic applications, their mechanisms of action, and the potential benefits and challenges associated with their use in cancer treatment. By understanding how bacterial therapies can influence immune responses and reshape the TME, we aim to provide insights into new avenues for enhancing the efficacy of cancer immunotherapy.

## 2. Background

The idea of stimulating the immune system with an infection to treat cancer originated from the development of Coley’s toxins by William Coley, a bone sarcoma surgeon, during the late 1800s and early 1900s [7]. Coley pioneered modern immunotherapy by administering a mixture of heat-killed Streptococcus bacteria and Serratia marcescens, known as Coley’s Toxins [8]. He observed that his patients experienced tumor regression, earning him the title “The Father of Immunotherapy” [9].

The application of Coley’s toxins, along with bacterial therapy as a whole, experienced a decline in the early 20th century due to inconsistencies in Coley’s work and other contributing factors, such as other leading bone sarcoma surgeons disavowing his work. In 1962, the FDA prohibited the use of this treatment for cancer. Nonetheless, Coley’s children contributed to changing public perception of their father’s methods through research and publications, thereby facilitating the subsequent acceptance of immunotherapy by the scientific community [10].

Building on this foundation of early immunotherapy research, subsequent decades saw the development and refinement of various bacterial-based approaches to cancer treatment. One of the most prominent examples of bacterial immunotherapy is the use of Bacillus Calmette–Guérin (BCG) in treating bladder cancer, which received FDA approval for this indication. Originally developed as a vaccine for tuberculosis (TB) in 1921, BCG’s potential as a cancer therapeutic emerged in 1929 when it was observed that patients with a history of TB exhibited lower cancer incidence compared to control groups. Interest in BCG as an anticancer agent surged in the 1970s, as numerous preclinical models demonstrated remarkable efficacy, with some studies reporting up to 90% success rates in treating melanoma. However, subsequent large-scale randomized clinical trials failed to consistently replicate these results across various cancer types, limiting the use of BCG to high-risk, non-muscle-invasive bladder cancer [11].

Despite these setbacks in broader applications, the potential of bacterial immunotherapy remained an area of interest, prompting researchers to explore new avenues and technologies. In recent years, advancements in gene engineering tools such as specialized animal models, Next Gen Sequencing (NGS), or CRISPR-Cas9 have helped revive bacterial immunotherapy as a viable alternative to chemotherapy and other standards of care [12]. NGS has been used to understand much larger sequences of DNA and their relation to cancer, while specialized animal models have been used to test and prove concepts [13]. Over time, these tools have brought back bacteria as a more advanced possible immunotherapy treatment. Immunotherapy uses bacteria to activate several factors, such as immunomodulatory agents, immune checkpoint inhibitors, or cytokines. Modern bacterial therapies have been engineered to activate these factors more efficiently to target various diseases, including cancer [14].

Research into targeting diseases with bacterial therapeutics has advanced in recent years [15]. For example, *Lactococcus lactis* was proven to be a useful treatment for cholera by converting simple carbohydrates into lactic acid, which would go on to harm *Vibrio cholera* inside the gut microbiome [16]. Another example is genetically engineered probiotics, which were found to increase gut health by producing antimicrobial substances such as bacteriocin [17]. Genetically engineered Bifidobacterium longum is being researched as a potential treatment for ulcerative colitis, an inflammatory disease affecting the colon. This approach utilizes the bacteria as a delivery system to distribute recombinant human manganese superoxide dismutase (rhMnSOD), a potent antioxidant enzyme with anti-inflammatory properties. In experimental mouse models, this engineered probiotic has shown promise in suppressing colitis by delivering rhMnSOD directly to the affected areas of the colon [18]. Other examples include Lactobacillus reuteri being used to lower fatty liver disease by delivering Interleukin-22 to the active site, or *E. coli* Nissle, being used to treat inflammatory bowel disease for its protection against other bacterial strains such as Salmonella, Yersinia enterocolitica, and Listeria pneumophila [19,20].

While these applications demonstrate the versatility of bacterial therapies in treating various conditions, the potential of engineered bacteria in oncology has garnered particular attention in recent years. Bacterial therapies are being actively explored for various cancers, with several notable examples in clinical trials. VNP20009, an attenuated strain of *Salmonella typhimurium*, entered phase I human trials in 1999 for patients with metastatic melanoma and renal cell carcinoma. While tumor colonization was observed at the highest tolerated dose, no significant antitumor effects were seen, indicating the need for further refinement [21,22]. Another promising approach involves Listeria-based vaccines targeting human papilloma virus (HPV)-related cancers, including cervical cancer. ADXS11-001, one such vaccine, has progressed through phase II and III studies [23]. The development of these therapies continues, with ongoing efforts to improve their efficacy and safety profiles [24]. *Clostridium novyi*-NT, an anaerobic bacterium, has also entered clinical trials for solid tumor malignancies. Phase I trials (NCT01118819 and NCT01924689) began earlier than 2019, evaluating its ability to germinate in and destroy hypoxic tumors [25,26] (Figure 1).

Given the versatility and effectiveness of genetically engineered bacteria, their application in cancer treatment holds tremendous promise. The burden of cancer in the United States remains significant, with projections indicating a continued high incidence and mortality rate. By the end of 2024, it is estimated that approximately 2,001,140 new cancer cases will be diagnosed across all sites in the United States, while cancer is projected to claim an estimated 611,720 lives during the same year [8]. From 1999 to 2016, the number of reported cases of cancer increased from 1.3 million to 1.7 million, with a peak at 1.8 million [27]. As the incidence of cancer increases, bacteria that can affect the TME’s immunosuppressive elements will be critical. These staggering statistics underscore the urgent need for innovative treatments, such as combination therapies that incorporate genetically engineered bacteria to boost the body’s immune response against cancer in conjunction with standard treatments like chemotherapy or checkpoint blockade. This approach brings exciting possibilities and hope to millions affected by this disease worldwide, representing a significant step forward in the fight against cancer.

## 3. Why Is TME Immunosuppressive?

In understanding the complexities of the tumor microenvironment (TME), it becomes evident that its immunosuppressive nature stems from several key factors. These include immune suppressor cells, the secretion of immunosuppressive cytokines, the expression of immune checkpoint molecules, and the hypoxic environment. Each of these components plays a critical role in shaping a hostile environment that supports tumor survival and hinders effective anticancer immune responses.

### 3.1. Presence of Immunosuppressive Cells

Immune suppressor cells such as myeloid-derived suppressor cells (MDSCs), regulatory T cells (Tregs), and tumor-associated macrophages (TAMs) play a role in the immune suppressive nature of the TME (Figure 2).

MDSCs: These are a diverse group of immature myeloid cells that actively suppress the immune system. This feature allows them to play a significant role in helping tumors evade immune responses, particularly in cancer. They function through a variety of mechanisms, such as the expression of NADPH oxidase (Nox2) proteins, S100A8/9, arginase 1 (Arg-1), and oxide synthase 2 (Nos2). They are divided into two different subtypes: polymorphonuclear MDSCs (PMN-MDSCs) and monocytic MDSCs (M-MDSCs) [28]. Although PMN-MDSCs are non-abundant in a healthy individual, they expand to various degrees in cancer patients [29]. In cases of inflammation and tumor progression, MDSCs can suppress the activity of CD4+ T cells, CD8+ T cells, and NK cells, thereby inhibiting the innate and adaptive immune responses [30,31,32]. For example, in breast cancer cells, MSDCs are recruited along with Tregs and type 3 macrophages, which downregulate antitumor immunity [33].

Tregs: A critical component of the immunosuppressive environment, this subset of CD4+ T cells is characterized by the expression of the transcription factor FoxP3 [34]. They are essential for maintaining immune homeostasis and preventing autoimmune responses [35]. In the context of cancer, Tregs are recruited to the TME, where they suppress effector T cells and other immune cells through mechanisms such as secreting immunosuppressive cytokines (e.g., IL-10, TGF-β), expressing immune checkpoint molecules (e.g., CTLA-4, PD-1), and altering metabolic pathways. Recent data show that Tregs also suppress bacteria-driven inflammation that promotes carcinogenesis in cancers like colorectal carcinoma (CRC), potentially benefiting the host. This dual role of Tregs helps explain why their frequency and function are linked to poor prognosis in some cancers but favorable outcomes in others [36]. Given their complex roles in both suppressing antitumor immunity and modulating inflammation, understanding Tregs’ functions within the TME is crucial for developing targeted immunotherapeutic strategies that can either harness their beneficial effects or mitigate their adverse impacts on cancer progression.

TAMs: Macrophages within the TME comprise diverse populations of myeloid cells that play critical roles in tumor progression, immune evasion, and metastasis. Increasing evidence indicates that TAMs contribute to tumor angiogenesis, enhance the motility of tumor cells, and facilitate extracellular matrix remodeling [37]. Based on their functional roles and marker expression, TAMs can be categorized into two main types, M1 and M2 [38,39]. However, TAMs exhibit significant phenotypic diversity and plasticity, with many subsets expressing a mix of M1 and M2 polarization markers [40,41].

**Figure 2 cancers-16-03810-f002:**
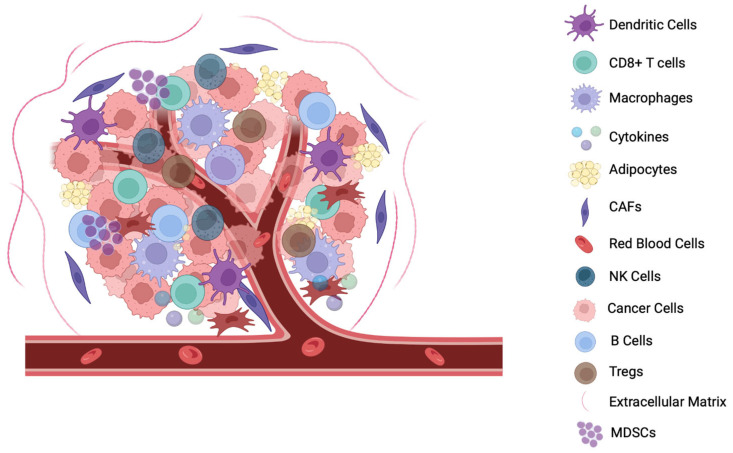
Components of the TME. The TME comprises a diverse collection of cancer cells, stromal cells, immune cells, the extracellular matrix (ECM), blood vessels, and various signaling molecules. Cancer-associated fibroblasts (CAFs), adipocytes, and the ECM provide structural support and promote tumor growth. Immune cells, such as tumor-associated macrophages (TAMs), dendritic cells, NK cells, CD8+ T cells, B cells, regulatory T cells (Tregs), and myeloid-derived suppressor cells (MDSCs), play complex roles, either supporting or inhibiting tumor progression. Blood vessels within the TME are often abnormal, contributing to hypoxia. Cytokines, chemokines, and growth factors facilitate cell communications, influencing tumor behavior and response to treatment [42]. This intricate interplay within the TME profoundly impacts cancer development, immune evasion, and therapeutic outcomes.

### 3.2. Secretion of Immunosuppressive Cytokines

Chronic inflammation can be found in the TME as it is mainly initiated by the presence of malignant cells. This can cause proinflammatory mediators like cytokines and chemokines to be elevated, thereby establishing crosstalk between various cells that can establish a tumor-supporting environment [43]. Interleukin (IL)-6, IL-17, IL-22, and IL-23 are certain cytokines essential to linking inflammation to tumorigenesis in colitis-associated colorectal cancer (CRC) mouse models [44,45,46]. Transforming growth factor-beta (TGF-β), IL-10, and vascular endothelial growth factor (VEGF) are some immunosuppressive cytokines that the cancer cells and the surrounding stromal cells secrete. It has been reported that cancer patients produce a large amount of VEGF, which is detectable in the serum of patients [47]. Increased VEGF expression leads to immune suppression in multiple ways, including reducing T cell infiltration into the TME, inhibiting dendritic cells, and promoting inhibitory cell types in the TME. In animal and human tumor models, it has been shown that the inhibition of VEGF significantly increases the number of tumor-infiltrating lymphocytes [48]. Like VEGF, the tumor produces increased TGF-β, which acts on the cytotoxic T lymphocytes (CTLs) and inhibits specific cytolytic gene products. TGF-β dysfunction mainly affects three main areas: evasion of immune surveillance, loss of growth inhibitory control, and acquisition of mutagenicity [49]. It has been shown that mice whose T cells have been rendered resistant to TGF-β through the production of a dominant-negative TGF-β receptor transgene can develop an immunological response, eradicate the tumor burden, and survive with TGF-β-producing tumors [50,51].

### 3.3. Expression of Immune Checkpoint Molecules

Immune checkpoint molecules play a crucial role in cancer treatment by regulating the immune system’s response to cancer cells. These molecules can activate or inhibit immune responses, influencing how effectively the body can target and destroy cancer cells. Key immune checkpoint molecules include PD-1, PD-L1, CTLA-4, CD28, LAG-3, and TIM-3 [52]. The expression of immune checkpoint markers can vary depending on the cancer. Nonetheless, several common high and low expression patterns of these markers are seen in various cancer types.

Below is a summary of important immunological checkpoint markers:

PD-1/PD-L1: Even though a full understanding of the mechanisms of the evasion of cancer cells from the immune system has remained elusive, one of the key signaling pathways involves the programmed death-1 (PD-1) receptor and its ligands PD-L1 and PD-L2. The results of multiple clinical studies with anti-PD-1 blockade monoclonal antibodies have validated that targeting the immune checkpoint of the host through PD-1 receptor blockade produces strong antitumor immune responses, which results in higher survival rates [53]. However, these checkpoint treatments are only beneficial for a subset of patients. By further understanding the immune system features within the TME that influence responses and resistance, it will be easier to identify which patients will benefit most from anti-PD-1 treatment alone. Furthermore, this understanding can uncover other immune factors that could be targeted along with PD-1 to improve outcomes [54].

CTLA-4/CD28: CTLA-4 is a protein receptor that functions as an immune checkpoint. It is expressed on the surface of T cells and is primarily involved in downregulating immune responses by ensuring that T cells do not become overactive. It is homologous to CD28 [55]. CTLA-4 competes with CD28 for binding to CD80/CD 86. When CTLA-4 binds to these molecules, it switches off antigen-presenting cells (APCs) [56] (Figure 3).

LAG-3: LAG-3 is encoded by the LAG-3 gene and is expressed on the cell surface of natural killer cells (NK cells) [58], B cells cells [59], tumor-infiltrating lymphocytes (TILs), and a subset of T cells [60], as well as dendritic cells (DCs) [61,62]. LAG-3 interacts specifically with tumor cells and antigen-presenting cells (APCs) via the major histocompatibility complex class II (MHC-II) [63,64], which triggers co-inhibitory signaling pathways and ultimately results in T cell alterations [65,66]. CTLA-4 and LAG-3 have comparable roles in immune regulation [67]. While CTLA-4 increases cell cycle arrest, decreases T cell receptor signaling, and inhibits T cell activation [61,67,68,69], LAG-3 influences T cell function differently; for instance, activated LAG-3−/− T cells exhibit increased mortality and extended cell cycle progression [70]. It has been reported that LAG-3 and PD-1/PD-L1 work cooperatively. Inhibiting PD-1 and LAG-3 increased helper T cells and B cells while activating CD4+ T cells, which also eliminated blood-stage malaria in mice [71].

TIM-3: A 2001 investigation of the genes linked to asthma susceptibility in congenic inbred mice led to the discovery of TIM-3, a member of the TIM gene family [72]. T cells (apart from Th2 cells) and other immune cells, including NK cells, macrophages, DCs, MDSCs, and mast cells, have all been shown to express TIM-3 [73,74]. High expression levels of TIM-3 are associated with T cell exhaustion, which is the progressive loss of T cell activity in a hierarchical fashion during long-term viral infections and tumor formation [75].

### 3.4. Hypoxic Environment

Hypoxia is a state of low oxygen concentration due to the lack of blood supply within the TME. This state is a hallmark feature of solid tumors and is a significant contributor to cancer progression, treatment resistance, and poor patient prognosis [76]. To cope with hypoxia, tumor cells develop resistance to low oxygen levels, marked by changes in signaling, gene expression, and metabolism [77]. These phenotypic changes help the cancer cells survive, proliferate, and ultimately invade distant sites [78]. Recent research has further elucidated the mechanisms by which hypoxia promotes tumor aggressiveness, highlighting the role of hypoxia-inducible factors (HIFs) in regulating these adaptive responses. Targeting hypoxic regions within tumors has become a focal point for developing novel therapeutic strategies aimed at disrupting these adaptive mechanisms and improving treatment outcomes [79,80].

One promising approach involves the use of bacterial agents in cancer therapy. By leveraging their natural ability to interact with the immune system, bacterial agents and their components can disrupt the immunosuppressive TME, particularly by targeting hypoxic regions and immune suppressor cells. Studies have demonstrated that engineered bacteria can be tailored to enhance immune detection and response, lysing tumor cells and thereby improving the efficacy of existing immunotherapies. This dual strategy not only alleviates the immunosuppressive conditions but also introduces a novel method to deliver therapeutic agents directly to the tumor site, marking a significant advancement in cancer treatment modalities [80,81,82].

## 4. Bacterial Agents in Immunomodulation

### 4.1. Bacterial Components

Bacterial components such as RNA, DNA, peptidoglycan, lipoteichoic acid, lipopolysaccharide, and flagellum can be recognized by pattern recognition receptors on neutrophils, dendritic cells, and macrophages. This recognition generates the appropriate immunological defense response. Lipopolysaccharide (LPS) found in Gram-negative bacteria’s outer membrane is a powerful immunogenic microbial-associated molecular pattern (MAMP) that predominantly binds to TLRs or on immune cell membranes [83]. Elevated LPS levels can cause amplification of IL-6, activation of the NF-κB and TLR pathways, and phosphorylation of STAT3 [84]. These activities promote dendritic cell (DC) maturity and immune cell division, strengthening antitumor immunity. Similarly, bacterial flagella can stimulate the innate immune response via TLR5 [85]. Gram-positive bacteria lacking LPS or flagella, such as *Bifidobacterium*, activate macrophages, natural killer (NK) cells, dendritic cells (DCs), and B lymphocytes through components like peptidoglycan, extracellular polysaccharides, and DNA [86]. These effector cells generate immune-active cytokines or chemicals such as IL-1, IL-6, IL-12, TNF-α, IFN-γ, and nitric oxide (Figure 4).

### 4.2. Bacterial Targeting in Tumors

Some bacteria, such as *Salmonella* and *E. coli*, utilize their flagellar motility to target cancer regions. These bacteria, including facultative and obligate anaerobes, find the TME to be nutrient-rich, providing an ideal setting for their growth and activity [88]. Within the tumor, engineered variants of these bacteria exert anticancer effects by activating both innate and adaptive immune responses against cancer cells. The therapeutic agents they produce include immunomodulators, cytotoxic proteins, angiogenesis modulators, small interfering RNAs, and prodrug-converting enzymes [6]. Toll-like receptors (TLRs), which play a crucial role in innate immunity as the body’s first line of defense against pathogens, are also involved. TLRs identify infections and stimulate the innate immune system, bridging the gap between initial pathogen recognition by innate immune cells and the subsequent activation of the adaptive immune response. They facilitate the connection between innate and adaptive immunity by regulating antigen-presenting cells and key cytokines [89].

Various mechanisms have been reported to explain how bacteria specifically target tumors: (a) bacteria are stimulated by conditions in the TME [90], (b) bacteria become trapped in tumors due to their disordered vasculature [91], and (c) bacteria are attracted to specific components of the TME [92]. Investigators found that injecting *Salmonella enterica* Serovar Typhimurium intravenously resulted in a considerable rise in TNF-α concentrations in the blood in an ectopic tumor model [93]. This disrupted blood arteries inside the tumors, resulting in bleeding and allowing bacteria to enter the tumor. Neutralizing TNF-α in tumor-bearing mice serum reduced blood input into tumors and delayed bacterial colonization. These findings highlight the crucial function of TNF-α and induced bleeding in the early phases of bacterial interactions with cancer tissue [93].

Bacteria can infiltrate tumor tissue through both passive and active pathways. In the passive pathway, bacteria get trapped in the disordered tumor vasculature and migrate within the tumor due to inflammation from a rapid surge in TNF-α [93]. The active pathway involves chemotaxis toward chemicals from dying tumor tissue and the low oxygen content in hypoxic tumors [94,95]. These pathways are not strain-dependent or mutually exclusive, allowing bacteria to selectively target tumors.

Low oxygen levels attract obligate anaerobes like *Clostridium* and facultative anaerobes like *Bifidobacterium* [96,97]. The deoxygenated tissue of tumors allows these anaerobes to accumulate in hypoxic zones [95]. Studies confirm this selectivity, as injecting *Clostridium* into tumor-bearing mice resulted in infection-related death only in those animals [94]. Facultative anaerobes like *E. coli* and *Salmonella typhimurium* use chemoreceptors to sense nutrient-rich conditions at tumor sites, causing them to accumulate on the tumor’s edges and within the tumor area [98] (Figure 5).

Bacteria localize inside the TME, interacting intricately with the tumor microenvironment, malignant cells, and bacterial cells. These interactions (Figure 5) facilitate tumor suppression by altering cytokines, chemokines, and immune cells infiltrating the tumor. Toxins from bacteria like *Salmonella*, *Listeria*, and *Clostridium* induce cytotoxicity in tumors by triggering autophagy or apoptosis [99,100,101,102]. *Salmonella* toxins upregulate Connexin 43 (Cx43), forming gap junctions between tumor and dendritic cells (DCs), facilitating tumor antigen cross-presentation to DCs [103,104]. DCs release IL-1β in response to tumor antigens and bacterial components, activating CD8+ T cell lymphocytes [104,105]. Bacterial flagellin activates TLR5, boosting the antitumor response of activated CD8+ T cells [85]. Perforin and granzyme proteins from activated CD8+ T cells [102,104] destroy tumor cells in both primary and metastatic sites. Flagellin and TLR5 signaling reduces CD4+, CD25+, and Tregs, enhancing the antitumor response [85]. A modified strain of *Brucella melitensis* lacking the vjbR gene stimulates macrophages to release IL-6, IL-1β, and TNF-α, activating CD8+ T cells [106]. Flagellin from *S. typhimurium* increases interferon-γ (IFN-γ) production by natural killer (NK) cells, which is essential for both adaptive and innate immunity [103]. *Listeria*-infected MDSCs transform into an immune-stimulating phenotype with increased IL-12 production, strengthening NK and CD8+ T cell responses [107]. Infections with *S. typhimurium* and *Clostridium* significantly increase neutrophils, which secrete more TNF-α and TRAIL, enhancing the immune response and tumor cell death [87,93,105,108]. When *Salmonella* damages cancer cells and bacterial components like flagellin and LPS are encountered, the macrophage inflammasome is activated, increasing IL-1β and TNF-α production in the tumor microenvironment [105,109].

### 4.3. Bacterial Engineering in Cancer Immunotherapy

Bacteria-based cancer immunotherapy has gained significant attention recently due to its innovative mechanisms and versatile applications in enhancing the body’s antitumor responses. Certain bacteria, such as *Brucella [106]*, *S.typhimurium* [87], *Bifidobacterium* [110], *Clostridium* [26,111], and *E. coli coli* [112], possess a unique ability to target and invade the core of tumors, thriving in the low-oxygen, high-nutrient environments that tumors provide. These bacteria are rich in pathogen-associated molecular patterns, which can activate immune cells even within the immunosuppressive tumor microenvironment, thereby boosting the immune system’s ability to detect and destroy cancer cells. However, when using bacteria to treat cancer, it is critical to minimize their harmful effects on the immune system while preserving their natural anticancer properties. Achieving this balance is challenging, but it has been accomplished with certain virulent strains by deleting key virulence genes, resulting in safe yet effective bacterial variants [106]. To dig deeper into how bacteria might help fight cancer, we describe the anticancer activities of various bacterial strains as well as their unique roles in boosting anti-tumor responses.

*A.* *Salmonella* 

The deletion of the msbB and purI genes in *S.typhimurium* developed the VNP20009 strain, which is commonly employed in anticancer research in tumor-bearing mice [113]. The deletion of the msbB gene in *Salmonella* dramatically decreases LPS-induced septic shock. Strains with lower virulence and increased therapeutic efficiency were produced by incorporating the LPS gene into the chromosome at the araBAD locus [21]. This method also downregulates genes linked with endotoxin, resulting in the development of a non-toxic *Salmonella* strain with increased therapeutic potential [114]. Furthermore, in the SL1344 strain, co-administration of Pimozide and sh-PD-1 boosted T cell infiltration, spleen CD4 + T Cells, CD8 + T cells, and NK cells, and decreased spleen Tregs in animal models. Additionally, the combined therapy enhanced survival and induced tumor death as compared to single monotherapies [115]. Also, *Salmonella* strains with relA and spoT gene mutations exhibit low toxicity due to a lack of ppGpp, which regulates toxin gene production. These strains have potent anticancer action via activation of the inflammasome (IPAF, NLRP3), which causes the release of different proinflammatory cytokines. It also reduced tumor development and metastasis in the HCT116 xenograft model. This therapy also activated M1-like macrophage polarization, resulting in the release of tumor-suppressive cytokines and nitric oxide [87]. These modified bacteria can reduce toxicity while increasing therapeutic efficiency by inducing anticancer immune responses and activating proinflammatory cytokines.

*B.* *Brucella* 

*Brucella* is a gram negative facultative intracellular bacterium [116] that uses effector proteins [117] o targets host signaling systems [118] to remodel host intracellular membrane trafficking systems [119] to establish an intracellular niche [120]. A live attenuated strain of *Brucella melitensis* [121], engineered with a deletion of the vjbR gene, plays a crucial role in regulating T cell function. This bacterium polarizes macrophages, leading to the release of proinflammatory cytokines such as IL-6, IL-1β, and TNF-α, which in turn activate CD8+ T cells [106], thereby enhancing the body’s ability to mount an effective antitumor response. This mutation not only reduces the virulence of the bacterium but also enhances its utility in cancer immunotherapy by selectively targeting tumor cells while minimizing damage to normal tissues [93]. In other work, the engineered variants of Brucella were shown to be useful for addressing autoimmune disease [106].

*C.* *E. coli* 

In recent studies, programmed *E. coli* Pir1+ was used to secrete a CD47-neutralizing nanobody (nb) using a synchronized lysis circuit (SLC), which initiates cell lysis based on the abundance of bacteria in the tumor. In mice, localized administration of CD47nb increased the activation of tumor-infiltrating T cells while dramatically inhibiting tumor development [122] In the follow-up study, *E. coli Nissle 1917* was engineered with SLC in a subsequent effort to generate nanobodies targeting PD-L1 and CTLA-4. In syngeneic animal models of the tumor microenvironment, PD-L1nb and CTLA-4nb released in situ enhanced T cell activation, increasing systemic memory T cell populations while delivering regression within established tumors. Most importantly, superior combination efficacy was observed using bacterial strains producing PD-L1nb, CTLA-4nb, and GM-CSF compared to its individual or combined immune checkpoint nanobody-producing strain [112]. These findings emphasize the promise of using modified bacteria for targeted immunotherapy, as well as the need to combine various immune-modulating drugs to obtain greater anticancer effects and better therapeutic outcomes.

*D.* *Lactobacillus* 

Recent developments in cancer immunotherapy have examined the ability of genetically engineered bacteria to convey tumor-associated antigens (TAAs) and trigger strong adaptive immune responses. This method uses bacteria’s capacity to present TAAs, resulting in targeted immune activation against cancer cells. A recent phase I/IIa clinical study evaluated an oral vaccination derived from Lactobacillus casei GLBL101c, which was modified to express the tumor-associated antigen HPV16 E7. This vaccination effectively reduced cervical intraepithelial neoplasia cancers in 70% of patients, lowering the severity from CIN3 to CIN1-2. The primed T cells that recognized the E7 antigen effectively targeted the cervical tumor and elicited a Th1 response [123].

*E.* *Bifidobacterium* 

The hypoxic tumor microenvironment is a key obstacle to successful cancer therapy, but new research has investigated the ability of engineered bacterial strains to target and control this milieu [6]. In one such study, a recombinant strain, *Bifidobacterium longum* C-CPE-PE23, was created to increase apoptosis and decrease tumor development, notably in triple-negative breast cancer (TNBC). This strain was designed to produce a recombinant toxin, including the CL-4-binding domain of C-CPE linked to PE23, a bacterial toxin with strong cytotoxicity against cancer cells. In vivo, experiments revealed that *B. longum* C-CPE-PE23 dramatically reduced tumor growth and increased apoptosis in a TNBC mouse model [110]. Another promising strategy in oral vaccination involves modifying *B. longum* to express the Wilms’ tumor 1 (WT1) protein. This vaccination inhibited WT1-expressing tumor development in mice by enhancing the cytotoxic activity of WT1 epitope-specific cytotoxic T cells. In a follow-up investigation, the recombinant bacterium displays WT1 protein. *B. longum* 420 expresses tumor cells demonstrating antitumor effect via mic splenocyte isolation and cytokine production. *B. longum* 420 significantly boosted CD4+ and CD8+ T cell infiltration into tumors, elevated systemic WT1-specific cytokine production, and improved the cytotoxic activity of WT1-epitope-specific cytotoxic T lymphocytes [124]. These advances highlight the intriguing significance of genetically modified bacteria in cancer immunotherapy by demonstrating their ability to efficiently display tumor-associated antigens and stimulate specific immune responses.

*F.* *Clostridium* 

Recent advances in bacterial-based cancer therapeutics focus on customizing *Clostridium* species to exploit hypoxic microenvironments within tumors. Engineered strains of *Clostridium*, such as *C. novyi*-NT and *C. butyricum*, are designed to thrive in low-oxygen areas of tumors [26,111]. These bacteria produce toxins that rupture tumor cell membranes and disrupt intracellular pathways [125]. Additionally, these engineered strains can recruit neutrophils, which produce TNF-related apoptosis-inducing Ligand (TRAIL), attracting CD8+ T lymphocytes, granulocytes, and macrophages [108,126].

*G.* *Listeria* 

*Listeria monocytogenes* has been the focus of recent studies, primarily targeting myeloid-derived suppressor cells (MDSCs) in the tumor microenvironment. When MDSCs take up *Listeria*, they can translocate to cancer cells through a cell-to-cell transport mechanism [107,108]. This method reduces tumors by directly killing invasive cancer cells and stimulating cytotoxic T cells specific to *Listeria* antigens [102]. Additionally, *Listeria* infection can induce certain MDSCs to adopt an immune-stimulatory phenotype, releasing interleukin-12 (IL-12), which enhances T cell and natural killer (NK) cell responses [107]. The bacteria also act directly on cancer cells by increasing intracellular calcium levels by activating NADP(+) oxidase and inducing ROS generation [102]. These findings demonstrate the potential of bacterial infections to overcome immune suppression, improve cancer treatment outcomes, and promote both direct cancer cell death and enhanced immune responses.

## 5. Immunomodulation of the TME

Immunomodulation of the TME leverages the body’s immune cells, including macrophages, dendritic cells, and T cells, to enhance antitumor responses. By using innovative strategies such as targeted delivery systems and bacterial components, researchers aim to recruit and activate these immune cells within the TME. This approach involves shifting immune cell phenotypes to create a more hostile tumor environment. TAMs can be targeted in various ways using bacteria and their contents to repolarize them to an antitumor state [127]. Similarly, tumor-associated neutrophils (TANs) have distinct transcriptomic profiles and phenotypes from naive neutrophils [128]. Hence, developing treatment methodologies that target this may prove pivotal. While establishing T cell immunity is critical in modulating the TME, it happens only with the assistance of antigen-presenting cells such as dendritic cells. Several research findings have reported that bacteria-based delivery systems lead to DC maturation [129]. The goal is to suppress tumor growth and improve the effectiveness of cancer therapies, offering new hope in the fight against cancer.

This section delves into how engineered bacteria and their components modulate the TME by specifically affecting these immune cells—macrophages, neutrophils, dendritic cells, and T cells. By leveraging the unique properties of engineered bacteria, it is possible to reprogram the TME from a tumor-supportive state to one that actively combats cancer, offering promising avenues for innovative cancer therapies.

### 5.1. Immunomodulation of the TME by Macrophages

Macrophages exhibit remarkable plasticity in their functions as they can change their phenotypes based on environmental cues. The M1 phenotype is activated in response to a pathogen, releasing cytokines to recruit other immune cells. Meanwhile, the M2 phenotype is generally anti-inflammatory as it is involved in tissue repair and immune regulation [130]. Innovative strategies to modulate the TME through targeted delivery systems are currently in development, enhancing the recruitment and activation of immune cells to suppress tumor growth effectively.

Innovative strategies involving targeted delivery systems are being developed to capitalize on the plasticity of macrophages in TME. Several studies have utilized bacteria or bacterial components as carriers for drug delivery into the TME. Qing et al. used *E. coli* OMVs combined with CaP nanoparticles to neutralize the acidic TME and promote M1 macrophage infiltration. This led to the production of cytokines such as IL-12, IL-6, TNF-α, and IFN-γ and the recruitment of CD3 and CD8 T cells, reducing the presence of Tregs [131]. Lim et al. co-administered oxaliplatin (OXA) with *E. coli* (DH5-alpha) to target tumors. Upon *E. coli* colonization in the TME, the delivery of OXA repolarized macrophages to the M1 phenotype and recruited APCs, leading to cytotoxic T cell recruitment and tumor suppression [132].

Another promising strategy targets the CD47 receptor on tumor cells. CD47 is a five-transmembrane receptor that is ubiquitously found in healthy cells. Its primary function is interacting with the SIRPα receptor on macrophages and dendritic cells to maintain self-tolerance. Many tumor cells upregulate CD47 expression to escape the immune system. Researchers investigated strategies to disarm the CD47 shield that tumors use to evade immune attacks by macrophages. Feng et al. designed antibody-coated nanoparticles (OMV-CD47nb) to target CD47 on cancer cells and TLRs on macrophages. This “dual adaptor” system triggered a switch towards tumor-killing M1 macrophages and recruited antigen-presenting cells, priming T cell immunity [133]. In another approach, Chowdhury et al. engineered bacteria (eSLC-CD47nb) to deliver a CD47-blocking nanobody directly into tumors. This localized therapy induced a broader antitumor response beyond the treated area (abscopal effect). It increased M1-like macrophages within tumors, enhanced antigen presentation, and stimulated the proliferation and activation of cytotoxic T cells [122].

TLRs are critical in recognizing pathogen and damage-associated molecular patterns (PAMPs and DAMPs) and activating innate immune responses. In one such study, researchers engineered *Salmonella* to target TLRs in the TME. FlaB protein activated TLR4, recruiting macrophages and reprogramming them from M2 to M1 via TLR5. This M1 switch boosted ROS, NO, and cytokine production, killing cancer cells. This bacteria-based approach offers a potential cancer therapy by manipulating macrophages within the TME [122].

### 5.2. Immunomodulation of the TME by Neutrophils

Neutrophils act as first responders in the innate immune system and initiate inflammatory responses to any pathogens through phagocytosis, degranulation, and the formation of neutrophil extracellular traps (NETs). The presence of tumor-associated neutrophils (TANs) in the tumor microenvironment has often garnered attention as their phenotype exhibits both antitumor and pro-tumor activities, similar to macrophages. In a study focused on the role of neutrophils at the crossroads of inflammation and cancer, nanopathogenoids (NPNs) loaded with cis-platin were coated with OMVs derived from *E. coli*. These NPNs mimicked pathogen-associated molecular patterns (PAMPs) and were phagocytosed by neutrophils, directing them to the tumor sites. The formation of neutrophil extracellular traps (NETs) released the NPNs into the TME, delivering the therapeutic payloads and contributing to tumor eradication [134]. Yam et al. used bacterial loads of *S. aureus* and *BCG* to induce inflammation in the TME that caused a change in neutrophil function to a cytotoxic one [135].

CRM197, a non-toxic Diphtheria toxin (DT) derivative, shares its immunological traits and binds to HB-EGF, a receptor often upregulated in cancer, potentially enhancing therapeutic efficacy. In the context of CRM197 administration, there is a notable surge in neutrophil activity, likely contributing to its observed antitumor effects. This involved neutrophil recruitment to tumor sites, inducing cytotoxicity on cancer cells, or altering the tumor microenvironment [136]. Furthermore, releasing neutrophil-derived molecules, such as cytokines and reactive oxygen species, may enhance the immune response against cancer cells.

*C. novyi*-NT spores preferentially activate in oxygen-depleted tumor regions, induce tumor cell breakdown, and prompt immune reactions. Neutrophils, pivotal in the early inflammatory cascade, release ROS and proteases, facilitating tumor destruction as their activation initiates broader immune responses, shaping treatment outcomes [126]. The interplay between bacteriolysis, angiogenesis, and immune activation dictates efficacy, with neutrophil involvement critical in determining tumor eradication or persistence.

In the ongoing search for safer treatments for bladder cancer, a recent study reveals the potential of benign *Clostridium CBM588*. Significant inhibition of bladder cancer growth akin to BCG’s efficacy but devoid of its adverse effects was noted with CBM588. The activation of TRAIL release from neutrophils (PMNs) by both CBM588 and BCG is noteworthy, indicating their vital role in antitumor immunity. Enhancing therapeutic outcomes could be achieved by combining CBM588 with TRAIL-sensitizing agents such as sulforaphane [108].

### 5.3. Immunomodulation of the TME by Dendritic Cells (DCs)

Dendritic cells (DCs) play a crucial role in activating the immune response to protect the body from harmful microorganisms and cancer. DCs are central to the efficacy of bacteria-based cancer immunotherapies, leveraging their unique role as professional antigen-presenting cells. Upon exposure to bacterial components such as lipopolysaccharides (LPS) and CpG motifs, DCs undergo maturation and upregulate surface molecules necessary for T cell activation, including MHC classes I and II, CD80, and CD86 [137]. These mature DCs present tumor antigens derived from bacterial treatments to naive T cells, thus initiating a robust adaptive immune response [138]. Additionally, DCs release proinflammatory cytokines such as IL-12, critical for polarizing T cells toward a Th1 response and promoting cytotoxic T-lymphocyte (CTL) activity, enhancing antitumor immunity [139,140]. These mechanisms collectively position DCs as pivotal orchestrators of immune responses in bacteria-based cancer therapies, driving innate and adaptive immunity to target and eliminate tumor cells.

By establishing close contact with tumor cells, dendritic cells not only activate the immune response but also serve as crucial antigen-presenting cells (APCs), bridging the innate and adaptive immune systems and playing a pivotal role in generating T cell memory to prevent future invasions by pathogens.

Lin et al. employed an attenuated *Salmonella* strain (VNP20009) induced with L-Arabinose to secrete GM-CSF and IL-7, stimulating DC maturation and T cell proliferation. This combination, when used with anti-PD1 antibodies, resulted in effective tumor eradication [141]. Another study utilized *Streptococcus mutans* as a carrier to deliver tetrahedral framework nucleic acids (tFNAs) attached to 5-fluorouracil (5-FU) and aptamer AS1411. The bacteria infiltrated the hypoxic regions of the TME and aggregated in tumor-specific biofilms, ensuring targeted drug delivery and recruitment of T cells and DCs [81]. Ultimately, several active mechanisms in the TME disrupt DC functions, leading to inadequate T cell activation and potentially inducing T cell tolerance to TAAs. The promising results of bacterial therapies in modulating immune responses underscore the need to evaluate their clinical safety and efficacy.

### 5.4. Immunomodulation of TME by T Cells

Cancer cells function through intrinsic and extrinsic mechanisms; their bidirectionally allows them to evade immune system regulation and antitumor immune responses. T cells are major specialized cells of the immune system that play a central role as mediators of cellular immune responses. The basic classification and primary functions of different T cell types are presented in Table 1 [142].

In the early stages of cancer, malignant cells fail to elicit a sufficient immune response, leading to improper activation of the innate immune system, which results in a weakened adaptive immune response. The TME T cells can attack the tumor, while others support its growth and development. CD8+ lymphocytes activate CD4+ Th1 cells, which release IFN-y and IL-2. Meanwhile, TH17 cells produce IL-17A, IL-17F, IL-21, and IL-22, which both promote tumor growth [143,144]. T cell responses to some tumors can be inhibited by the intervention of two molecules: cytotoxic T lymphocyte-associated antigen-4 (CTLA-4) and programmed cell death protein 1 (PD-1) [145].

Alternatively, some tumors express Fas ligand (FasL), which binds to the Fas receptor on the leukocytes attacking the tumor, producing its apoptotic death [146]. Finally, tumor and TME cell products have the capacity to suppress antitumor immune responses. An example is TGF-β, a cytokine that can inhibit the proliferation and effector functions of lymphocytes and macrophages. It promotes the conversion of CD4+ T lymphocytes into regulatory T cells, which have an immunosuppressive effect [147].

As discussed, bacteria therapy can induce the production of cytokines in tumor cells that can attract and improve the antitumor activity of various lymphocytes. *Salmonella* Typhimurium has been used in immunotherapies in murine trials, with significant tumor reduction, resulting from the local expression of bacteria or the expression of immune system-stimulating molecules on tumor cells IL-18, CCL21, LIGHT, or the Fas ligand [148]. *Salmonella* flagellin has also been implicated in the reductions of CD4 + CD25+ T cells through TLR5 in the tumor microenvironment [149] and the induction of PD-L1 expression in dendritic cells and CD4+ cells [150,151].

Similarly, Garbatri et al. engineered *E. coli Nissle 1917* (EcN), a bacteria modified to express PD-L1 and CTLA-4 antagonists by transforming it with high-copy plasmid-carrying sequences for PD-L1 and CTLA-4 nanobodies. SLC was used to release the antagonists to maximize therapeutic efficacy. In multiple mouse models, intra-tumoral injections of the engineered EcN led to either partial or complete tumor regression. Immunophenotyping revealed an increase in intra-tumoral activated CD8+ T cells and conventional CD4+ T cells, along with a decrease in Tregs, indicating a shift from an immunosuppressive to a responsive tumor microenvironment and triggering a robust adaptive immune response [112].

## 6. Clinical Evidence for the Safety and Efficacy of Bacterial Therapy for Cancer

The clinical exploration of engineered bacterial agents in cancer therapy presents a nuanced understanding of both promises and limitations. Clinical trials are a testament to improving patient care by rigorously testing new interventions, albeit in a controlled setting. Phase I trials primarily focus on safety, involving a small number of participants to determine the maximum tolerated dose and initial safety profile of a new treatment. Phase II trials expand on safety assessments with larger patient populations and begin to evaluate preliminary efficacy measures, typically lasting longer than Phase I studies.

Safety was demonstrated at certain doses in a previously mentioned phase I study that administered attenuated *Salmonella* Typhimurium (VNP20009) intravenously to patients with metastatic melanoma. However, its efficacy in tumor colonization and antitumor activity remained limited. Only three out of twenty-five patients showed tumor colonization, receiving doses of at least 3 × 10^8^ cfu/m^2^. This limited colonization could be attributed to the rapid clearance of VNP20009 from the bloodstream, suggesting that while bacterial safety can be maintained, its persistence and interaction with the tumor microenvironment need further enhancement [7]. Additionally, similar observations (poor colonization and lack of an antitumor response) were reported when a live strain was used [152]. Since this was a phase I study, the aim was to look at the safety of the treatment modality, observe any adverse events, and determine the MTD.

Similarly, VXM01, an orally applied attenuated *Salmonella* Typhi strain, has been evaluated in a phase I trial for its anti-angiogenic effect via VEGFR2 targeting rather than a direct antitumor response. Notably, patients who exhibited reduced or stable tumor perfusion had elevated levels of VEGFR2-specific T effector cells. This suggests that continuous boosting might be required to maintain high levels of these T cells, especially in patients with advanced-stage cancers. Further trials with larger sample sizes and patient stratification based on tumor burden could improve our understanding of its efficacy [153].

Another promising bacterial candidate was explored in a phase I study targeting invasive cervical carcinoma using *Listeria monocytogenes* (Lm), which secretes HPV-16 E7 antigen. This live-attenuated strain increased the median survival rate by 347 days, a significant improvement compared to the historical median survival of 6–7 months. One patient showed an E7-specific T cell response, highlighting the potential of this approach to induce an immune response against HPV-associated cancers. However, larger cohorts and rigorous controls are essential to validate these findings, particularly to assess immunogenicity and clinical efficacy [154].

In a phase II trial of another attenuated *Listeria monocytogenes* strain, Axalimogene filolisbac (ADXS11-001) was tested in patients with metastatic cervical cancer. Despite two treatment arms (monotherapy and combination with cisplatin), the overall survival benefit was similar between the two, with no added advantage from the combination therapy. The study authors noted that including a cisplatin-only arm would have offered better insight into the adverse events and interactions between therapies, underscoring the importance of thoughtful trial design in multi-modal treatment studies [155].

A phase II study in metastatic pancreatic cancer evaluated the combination of cyclophosphamide, GVAX pancreas vaccine, and CRS-207 (*Listeria monocytogenes* expressing mesothelin). The addition of nivolumab (an anti-PD-1 immunotherapy drug) significantly improved overall survival, with a 56% improvement in the nivolumab-treated arm compared to Cy/GVAX alone. This finding emphasizes the potential for synergy between bacterial vaccines and immune checkpoint inhibitors in enhancing antitumor responses, particularly in hard-to-treat cancers like pancreatic cancer [156].

The clinical application of engineered bacterial agents in cancer therapy demonstrates both promise and limitations. Phase I and phase II trials highlight the safety and tolerability of these agents, though their efficacy in tumor colonization and antitumor responses remains inconsistent. While attenuated bacterial strains, such as *Salmonella* Typhimurium, VXM01, and *Listeria monocytogenes*, have shown the ability to elicit immune responses and, in some cases, modest survival benefits, challenges persist in optimizing bacterial persistence, tumor targeting, and enhancing immune responses. The integration of bacterial agents with existing immunotherapies, such as checkpoint inhibitors, has shown potential in improving outcomes, as seen in trials involving pancreatic and cervical cancers. However, further research is necessary to address key limitations, including refining trial designs, expanding patient cohorts, and improving bacterial interactions with the tumor microenvironment to fully harness their therapeutic potential.

## 7. Conclusions

Engineered bacteria hold immense promise as a novel approach to cancer therapy, offering a unique advantage in targeting the TME and enhancing immune responses. Unlike conventional treatments, which often lead to relapse [157], and systemic toxicity, bacterial-based therapies can precisely modulate immune responses and address the challenges posed by hypoxic tumor regions. Studies such as the use of *Clostridium novyi*-NT combined with radiation therapies demonstrate the potential of bacterial therapeutics to significantly improve treatment outcomes, including achieving complete remission in preclinical models. These bacteria enhance the efficacy of treatments by exploiting the unique conditions within tumors, such as hypoxia, which standard therapies often fail to address.

Furthermore, advancements in genetic engineering, particularly with CRISPR-Cas9, have allowed for the development of highly specialized bacterial strains capable of targeting specific cancer cells and modifying the TME. This enables better immune cell recruitment and function, thereby supporting tumor destruction. The integration of bacterial components with immune checkpoint inhibitors also presents a promising avenue for enhancing the immune system’s ability to fight cancer, further underscoring the potential for bacterial therapies to complement existing treatments.

Despite this progress, there remain challenges that must be resolved before these therapies can be widely adopted in clinical practice. Ensuring the long-term safety of engineered bacteria, minimizing off-target effects, and optimizing delivery methods are crucial areas that require further investigation. Moreover, the need for rigorous preclinical and clinical testing cannot be overstated, as it is essential to establish both the efficacy and safety of these innovative therapies.

In conclusion, engineered bacterial therapeutics represent a transformative frontier in cancer treatment, with the potential to overcome the limitations of conventional therapies. Ongoing research, coupled with advancements in bacterial engineering and combination strategies, will be critical in realizing their full potential. With continued innovation, bacterial-based therapies could significantly improve patient outcomes and offer new hope in the fight against cancer.

## Figures and Tables

**Figure 1 cancers-16-03810-f001:**
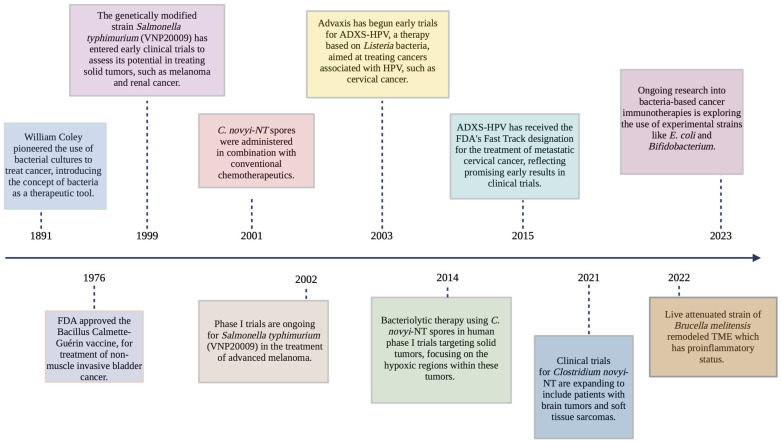
Advancements in bacterial-based therapies for cancer treatment.

**Figure 3 cancers-16-03810-f003:**
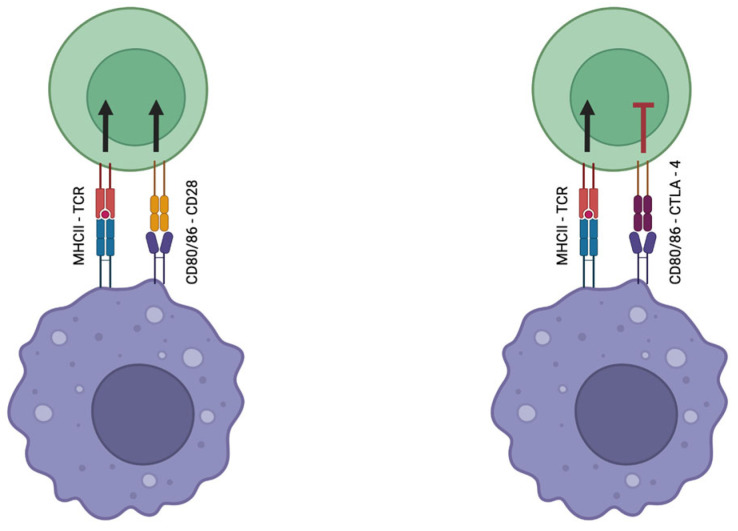
CTLA-4 and CD28 interaction with CD80/86. CTLA-4 competes with the costimulatory molecule CD28 for the CD80/86 ligands, where it has a higher affinity and avidity [55,57]. Because CD80 and CD86 both use CD28 to provide a positive costimulatory signal, CTLA-4’s function in competitively inhibiting CD28 is crucial for reducing T cell activation and adjusting the immunological response [57].

**Figure 4 cancers-16-03810-f004:**
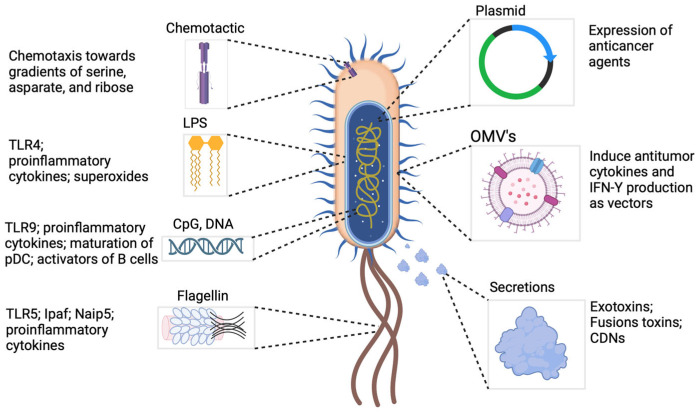
Representation of bacteria and bacterial components. Bacterial agents used in cancer therapy are intended to boost the immune system, allowing it to better detect and fight cancer cells. These agents are important in cancer therapy because of their ability to interact with the immune system, disrupt the immunosuppressive tumor microenvironment, and lyse tumor cells, thereby improving the immune response and the efficacy of existing immunotherapies and providing a promising avenue for novel cancer treatments. In recent years, advancements in technology and the attenuation of pathogenic strains have led researchers to concentrate on biochemical and molecular techniques to manipulate bacteria in the fight against cancer. These bacteria can be engineered to selectively target tumors and deliver anticancer medications, proteins, antibodies, enzymes, antigens, and cytokines straight to the cancer cells, thanks to developments in synthetic biology and genetic engineering [87].

**Figure 5 cancers-16-03810-f005:**
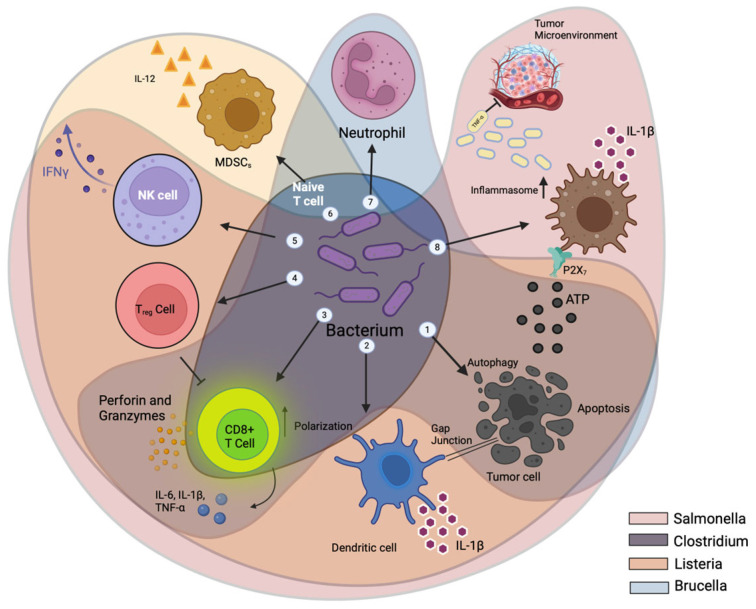
The mechanism by which bacteria target tumors. Bacterial toxins and components such as those from *Salmonella*, *Listeria*, *Clostridium*, and *Brucella melitensis* induce tumor cytotoxicity by triggering autophagy, apoptosis, and immune responses. These bacteria enhance CD8+ T cell activation, reduce regulatory T cells, and promote cytokine production (e.g., IL-1β, TNF-α, IFN-γ), ultimately boosting antitumor immunity through various mechanisms, including gap junction formation, inflammasome activation, and increased neutrophil activity.

**Table 1 cancers-16-03810-t001:** Classification of T cells.

**Cytotoxic T cells (CD8+ lymphocytes)**	Detect peptides presented by MHC class I molecules and destroy infected cells.
**Helper T cells (CD4+ lymphocytes)**	Detect peptides presented by MHC class II molecules; activate other cells of the immune system through the secretion of cytokines.
**Regulatory T cells**	Suppress immunity at the end of the immune reaction and maintain tolerance to autoantigens.
**Memory T cells**	Generated after the activation of T lymphocytes and will respond to new exposures to the same microorganism.

## Data Availability

The data generated in this present study are included in the figures and/or tables of this article.

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
