# Peer review of "Harnessing Bacterial Agents to Modulate the Tumor Microenvironment and Enhance Cancer Immunotherapy"

_cancers, 2024, doi:10.3390/cancers16223810_

Round 1
Reviewer 1 Report
Comments and Suggestions for Authors
Despite the remarkable successes of immune checkpoint blockades (ICBs) using anti-PD-1 and anti-CTLA-4 antibodies, the efficacy of ICB immunotherapy is currently limited. The tumor microenvironment (TME) is a complex entity that results in cellular, anatomical, and molecular barriers to anti-tumor immunity. The previous studies indicated that the microbiota and bacterial-based therapy contributed to the anti-tumor immune responses. In this manuscript entitled “Harnessing Bacterial Agents to Modulate the Tumor Microenvironment and Enhance Cancer Immunotherapy”, the authors comprehensively summarized the suppressive cells and molecules as well as the ER stress condition in the TME and how engineered bacterial agents regulate the immune cells and enhance cancer immunotherapy. This manuscript is well written and is an important topic. There are a few minor issues to be resolved to merit publication in Cancers.
1. It would be helpful if the authors could provide a list or table that allows readers to have a clear understanding of the progress of bacterial-based therapies in cancer treatment.
2. It might be more logical if the authors could switch section 4 with section 5.
3. It is better to use the words consistently, such as PD-L1 vs. PDL-1, anti-tumor vs. antitumor, TIM3 vs. TIM-3.
4. There are some typographical and font issues. The fonts are different from lines 105-117 and 336-343.
Author Response
Comment #1:
It would be helpful if the authors could provide a list or table that allows readers to have a clear understanding of the progress of bacterial-based therapies in cancer treatment.
Response:
We agree that a visual summary would enhance the clarity of the manuscript. In response, we have created Figure 1, which presents a comprehensive overview of key bacterial-based therapies in cancer treatment. This figure has been included in the revised manuscript on page 4.
Comment #2:
It might be more logical if the authors could switch section 4 with section 5.
Response:
After careful consideration, we have chosen to retain the current structure. We believe it is clearer to present background information on bacterial mechanisms first, which lays a solid foundation before discussing immune cell interactions. This structure helps readers better understand the subsequent discussion of immune cell involvement in bacterial-based cancer therapies.
Comment #3:
It is better to use the words consistently, such as PD-L1 vs. PDL-1, anti-tumor vs. antitumor, TIM3 vs. TIM-3.
Response:
We have thoroughly reviewed the manuscript to ensure consistent terminology throughout. The terms have been standardized as follows: "PD-L1," "antitumor," and "TIM-3." These changes are reflected in the revised manuscript.
Comment #4:
There are some typographical and font issues. The fonts are different from lines 105-117 and 336-343.
Response:
The typographical and font inconsistencies have been corrected. The font throughout the manuscript is now uniform, including lines 105-117 and 336-343.
Reviewer 2 Report
Comments and Suggestions for Authors
The manuscript provides a thorough overview of the historical evolution of bacterial therapy, starting with Coley’s toxins and exploring how advances in gene engineering have revived this approach. It emphasizes the complexity of the tumor microenvironment (TME), which includes various immune and non-immune components that hinder immune cells from effectively targeting tumors. The review discusses how bacterial agents influence different immune cell populations and highlights strains currently being explored in cancer therapy. It also addresses the potential benefits and challenges of bacterial therapies, with a special focus on developing combination therapies to enhance cancer treatment outcomes.
Strengths:
- The review offers a detailed and insightful analysis of the role the TME plays in cancer progression and immune evasion. It effectively explains how immune cells, such as myeloid-derived suppressor cells (MDSCs), regulatory T cells (Tregs), and tumor-associated macrophages (TAMs), contribute to the immunosuppressive landscape, adding depth to the discussion.
- By tracing the roots of bacterial immunotherapy back to Coley’s toxins, the paper adds valuable historical context, helping readers understand the progression of bacterial therapies over time.
- The manuscript thoroughly examines how bacterial agents modulate the TME to enhance immunotherapy, highlighting specific mechanisms through which these agents interact with immune cells. This analysis provides useful insights into how bacterial therapies can be optimized for improved cancer treatment outcomes.
- The paper’s focus on the clinical relevance of bacterial therapies, particularly in enhancing the effectiveness of immunotherapy amid rising cancer incidence, makes the content timely and applicable to ongoing research.
Areas for Improvement:
- Incorporating more clinical trial results or case studies would strengthen the manuscript by showcasing real-world applications of bacterial therapies in cancer treatment. Including examples from pre-clinical and clinical studies involving bacterial agents in immunotherapy would add further credibility.
- The discussion of challenges, such as the dual role of bacterial agents and potential side effects, could be expanded to cover a wider range of risks, ethical considerations, and mitigation strategies. This would provide a more balanced view of the therapy’s potential hurdles.
- Expanding the section on future research directions could offer readers a clearer path forward, moving beyond summarizing current knowledge to propose innovative and actionable research ideas for advancing bacterial therapies. This would further enhance the manuscript's impact and forward-looking relevance.
Author Response
Comment:
Incorporating more clinical trial results or case studies would strengthen the manuscript by showcasing real-world applications of bacterial therapies in cancer treatment. Including examples from pre-clinical and clinical studies involving bacterial agents in immunotherapy would add further credibility. The discussion of challenges, such as the dual role of bacterial agents and potential side effects, could be expanded to cover a wider range of risks, ethical considerations, and mitigation strategies. Expanding the section on future research directions could offer readers a clearer path forward, proposing innovative and actionable ideas to advance bacterial therapies.
Response:
In response, we have added a new section (Section 6) that includes clinical trial results and case studies, demonstrating the real-world applications of bacterial therapies in cancer treatment. Additionally, we have expanded our discussion of the challenges and risks associated with bacterial therapies, as well as mitigation strategies. We also enhanced the section on future research directions, outlining innovative areas for further investigation.
Reviewer 3 Report
Comments and Suggestions for Authors
In this manuscript, the authors aim to review current progress on utilizing bacteria to modify the tumor microenvironment (TME) and improve immunotherapy outcomes. While the topic is interesting and the review is comprehensive, there are several concerns that need to be addressed:
1. The manuscript includes a substantial amount of basic immunology information that can be omitted to make the review more focused. For example, Figure 5 on tumor regulation of dendritic cells (DCs) is not discussed in the main text, and no bacteria-based treatments target the specific mechanism shown in the figure. Therefore, Figure 5 should be removed. Similarly, Table 1 should be deleted.
2. The introduction on checkpoint inhibitors should be concise and to the point.
3. Several citations are either missing or incorrectly cited: Reference #78 is placed in the wrong place, and Reference #6 cannot be found in the literature. Citations are missing for multiple statements, such as the one in line 499: “Several research findings have reported that bacteria-based delivery systems lead to DC maturation,” and the one in line 322: “Various mechanisms were reported to demonstrate the specificity of bacteria targeting tumors.”
4. The formatting of references is inconsistent- some include page numbers, while others do not.
5. The discussion section seems unrelated to the main topic and could either be removed or integrated into other sections.
6. The abstract format is unusual and does not concisely summarize the content.
Comments on the Quality of English Language
There are some format and grammar issues.
Author Response
Comment #1:
The manuscript includes a substantial amount of basic immunology information that can be omitted to make the review more focused. For example, Figure 5 on tumor regulation of dendritic cells (DCs) is not discussed in the main text, and no bacteria-based treatments target the specific mechanism shown in the figure. Therefore, Figure 5 should be removed. Similarly, Table 1 should be deleted.
Response:
We have removed Figure 5 as suggested to streamline the manuscript and maintain focus. However, we have retained Table 1 because it provides useful context for readers less familiar with immunology. We believe it adds value by offering a brief primer on key concepts related to bacterial therapies.
Comment #2:
The introduction of checkpoint inhibitors should be concise and to the point.
Response:
While we understand the need for conciseness, we have chosen to retain the current level of detail in the introduction, as it provides essential context for readers who may not have a strong background in checkpoint inhibitors or their relevance to bacterial therapies in cancer treatment.
Comment #3:
Several citations are either missing or incorrectly cited: Reference #78 is placed in the wrong place, and Reference #6 cannot be found in the literature. Citations are missing for statements such as the one in line 499: “Several research findings have reported that bacteria-based delivery systems lead to DC maturation,” and the one in line 322: “Various mechanisms were reported to demonstrate the specificity of bacteria targeting tumors.”
Response:
We have corrected the placement of Reference #78 and verified that Reference #6 is accurately cited in the literature on page 2, line 77. Additionally, we have added the necessary citations for the statements on lines 499 and 322 to ensure proper attribution.
Comment #4:
The formatting of references is inconsistent – some include page numbers, while others do not.
Response:
The reference list has been revised using the MDPI citation style in EndNote to ensure consistency in formatting.
Comment #5:
The discussion section seems unrelated to the main topic and could either be removed or integrated into other sections.
Response:
We have removed the original discussion section and integrated relevant points into other sections of the manuscript. We have also expanded the section on clinical trials to enhance the relevance and flow of the manuscript.
Comment #6:
The abstract format is unusual and does not concisely summarize the content.
Response:
We have revised the abstract to improve its format and ensure that it concisely summarizes the main points of the manuscript, in line with the journal’s guidelines.
Round 2
Reviewer 3 Report
Comments and Suggestions for Authors
The author has adequately addressed the concerns. I would recommend accepting it for publication.
4o